# Olfactory Dysfunction in Parkinson's Disease, Its Functional and Neuroanatomical Correlates

Gabriel Torres-Pasillas [1], Donají Chi-Castañeda [2], Porfirio Carrillo-Castilla [3], Gerardo Marín [4], María Elena Hernández-Aguilar [2], Gonzalo Emiliano Aranda-Abreu [2], Jorge Manzo [2] and Luis I. García [2,*]

1 Doctorado en Investigaciones Cerebrales, Universidad Veracruzana, Xalapa 91190, Mexico; zs20022668@estudiantes.uv.mx
2 Instituto de Investigaciones Cerebrales, Universidad Veracruzana, Xalapa 91190, Mexico
3 Instituto de Neuroetología, Universidad Veracruzana, Xalapa 91190, Mexico
4 Neural Dynamics and Modulation Lab, Cleveland Clinic, Cleveland, OH 44195, USA
* Correspondence: luisgarcia@uv.mx

**Abstract:** Parkinson's disease (PD) is known for its motor alterations, but the importance of non-motor symptoms (NMSs), such as olfactory dysfunction (OD), is increasingly recognized. OD may manifest during the prodromal period of the disease, even before motor symptoms appear. Therefore, it is suggested that this symptom could be considered a marker of PD. This article briefly describes PD, the evolution of the knowledge about OD in PD, the prevalence of this NMS and its role in diagnosis and as a marker of PD progression, the assessment of olfaction in patients with PD, the role of α-synuclein and its aggregates in the pathophysiology of PD, and then describes some functional, morphological, and histological alterations observed in different structures related to the olfactory system, such as the olfactory epithelium, olfactory bulb, anterior olfactory nucleus, olfactory tract, piriform cortex, hippocampus, orbitofrontal cortex, and amygdala. In addition, considering the growing evidence that suggests that the cerebellum is also involved in the olfactory system, it has also been included in this work. Comprehending the existing functional and neuroanatomical alterations in PD could be relevant for a better understanding of the mechanisms behind OD in patients with this neurodegenerative disorder.

**Keywords:** α-synuclein; assessment of the olfactory system; non-motor symptoms; olfactory dysfunction; Parkinson's disease





## 1. Introduction

Parkinson's disease (PD) is a neurodegenerative disorder characterized by four cardinal motor signs: resting tremor, rigidity, bradykinesia, and postural instability [1,2]. While its motor alterations are extensively described, it has been observed that it also presents a series of Non-Motor Symptoms (NMSs) that often precede clinical motor manifestations. Those NMSs are mood and sleep disorders, mainly rapid eye movement (REM) sleep behavior disorder (RBD), dysautonomia, and olfactory dysfunction (OD). Since these NMSs arise before the onset of motor symptoms, this period is called the premotor phase of PD [2,3].

OD is one of the earliest features of PD and has been found to occur even four years before the diagnosis of this neurodegenerative disorder [4]. This affects the patient's quality of life, considering that the olfactory system plays a crucial role in influencing food flavors, detecting whether they are spoiled, and identifying harmful volatile compounds. Therefore, hyposmia, defined as a decrease in the sense of smell, and anosmia, defined as a total loss of smell, can significantly affect the well-being of the affected individual [5]. It is worth noting that approximately 68% of PD patients are unaware of the severity of their olfactory loss [6].

Therefore, gaining a comprehensive understanding of OD and the functional and anatomical alterations in the brain structures involved in the olfactory system in PD patients is crucial to improving early diagnosis and treatment. For this reason, this study aims to review the relevant literature addressing OD in PD patients. This article describes some functional, histological, and morphological alterations in the peripheral olfactory structures and in the primary and secondary olfactory cortices that could contribute to developing these NMSs in PD.

## 2. Parkinson's Disease

Parkinson's disease is the second most prevalent neurodegenerative disorder worldwide after Alzheimer's. It is estimated that there are 10 million people with PD globally [7]. The disease has a higher prevalence in men than in women, typically occurring between the ages of 65 to 70. Although it can occur in individuals under 40, it represents only 5% of cases [8]. The etiology is not entirely defined, but it could be multifactorial, resulting from interactions between genetic and environmental factors. Patients with an implicated gene are said to have familial or hereditary PD, while those with no known specific cause are said to have idiopathic Parkinson's disease (IPD) [7]. Approximately 10–15% of all patients diagnosed with PD have familial PD, which is characterized by mutations in one of the following genes: *SNCA, LRRK2, GBA, VPS35, PINK1, PARK7*, and *PARK2* [7].

Regarding environmental factors, an association has been identified between PD and drug use, such as amphetamine, methamphetamine, or cocaine use, exposure to pesticides and heavy metals (such as iron, copper, manganese, lead, and mercury), exposure to solvents, or the entry of a virus [7,9]. A meta-analysis of 104 studies showed a clear association between the risk of PD and exposure to solvents and pesticides, including the herbicide paraquat and the fungicide mancozeb [10]. The results indicate that people exposed to these substances have twice the risk of developing this disease compared to those who are not.

The diagnosis of PD comprises a medical history and neurological evaluation in order to identify the presence of the four cardinal signs of the disease [1]. Different tests, such as Dopamine Transporter Single-Photon Emission Computed Tomography (DAT-SPECT), F-fluorodopa PET, transcranial ultrasound, and genetic tests, can complement the neurological evaluation [11]. In addition, the prescription of dopaminergic medications such as Levodopa and an evaluation of their response can be helpful [7]. This is because other Parkinsonian syndromes, such as Progressive Supranuclear Palsy (PSP) and Multiple System Atrophy (MSA), do not respond to these dopaminergic drug therapies [12].

In addition, NMSs, such as cognitive deficits, apathy, gastrointestinal dysfunction, cardiovascular problems, psychological disorders such as depression or anxiety, and sleeping problems, mainly RBD, can be considered for the diagnosis [13,14]. PD patients may also have sensory deficits, including visual difficulties, altered pain processing, and OD [13,15–17].

## 3. A Brief Review of the Evolution of Knowledge about Olfactory Dysfunction in Parkinson's Disease

In 1975, Ansari and Johnson documented that 10 out of 22 patients with PD had OD [18]. The study's authors suggested that this condition might be related to an alteration in the dopaminergic system in the olfactory bulb (OB). Considering the genetic, neurophysiological, and pathological evidence that PD patients develop OD, in 1999, Hawkes and colleagues proposed that PD could be a primary olfactory disorder [19].

In 2003, a study by Braak and colleagues provided a possible explanation for OD in PD patients. Examining the progression of the dissemination of $\alpha$-synuclein pathology in postmortem brains, they found that in addition to the dorsal motor nuclei of the vagus (DMV) and glossopharyngeal nerves, the OB and anterior olfactory nucleus (AON) were among the first sites at which $\alpha$-synuclein pathology is found in PD patients [20].

In 2004, Huisman and colleagues studied the OBs of 10 PD patients and 10 controls. They identified a 100% increase in dopaminergic cells in the patients, suggesting that dopamine could be responsible for OD in PD [21]. In 2008, the same authors conducted a study involving 20 PD patients and 19 controls. However, their findings differed, as they only observed a significant increase in these cells in females but not males. This suggests that OD cannot be solely explained by dopaminergic alterations in this brain structure, and other mechanisms must be involved [22].

In 2007, Hawkes and colleagues proposed the "dual-hit hypothesis for Parkinson's disease" [23] based on the findings of Braak and colleagues [20]. This hypothesis postulates that a neurotropic pathogen, such as a virus or toxins, enters the brain via two distinct pathways: the nasal route and the enteric nervous system plexuses. In the first pathway, the pathogen causes damage to the OB and then spreads toward the temporal lobe. In the second pathway, the pathogen propagates retrogradely from the enteric nervous system plexuses through the vagus nerve to reach the DMV in the brainstem. From there, it can spread to the midbrain, causing damage to the Substantia Nigra pars compacta (SNc), responsible for the motor symptoms characteristic of PD.

In 2008, Doty reviewed the plausibility of the olfactory vector hypothesis of neurodegenerative disease, which aligns with the dual-impact hypothesis for PD, suggesting that a xenobiotic agent enters the brain via the olfactory mucosa and then spreads to other brain structures [9]. This hypothesis also applies to AD and arises from Roberts' proposal (1986) that the disease could be caused by an agent that enters via this route, such as aluminosilicates [24].

Although several previous studies suggested the possibility that OD precedes motor symptoms in PD patients, a study that provided more significant evidence in this regard was carried out by Ross and colleagues and published in 2008. This study assessed and followed the olfactory function of 2267 men without clinical PD for eight years. During this period, 35 cases were diagnosed with PD. It was found that those who had scored lower on the smell identification test had a higher risk of developing PD within the following four years [4].

Considering the broad evidence that most PD patients develop OD, in 2015, it was integrated as a symptom within the clinical diagnostic criteria for Parkinson's disease by the Movement Disorder Society (MDS) [25]. In 2022, a study by Borghammer and colleagues provided evidence that does not support the dual-hit hypothesis of PD; after analyzing a dataset of 302 postmortem brains with Lewy pathology, they found evidence of cases in which the lower brainstem or peripheral autonomic nervous system was affected, but without Lewy pathology in the OB [26]. Their findings support the α-Synuclein Origin site and Connectome (SOC) model proposed by the same author [27].

The SOC model proposes that α-synuclein pathology can originate in two sites: (1) the OB or amygdala and (2) in the enteric nervous system. However, unlike the dual-hit hypothesis of PD, α-synuclein pathology does not co-occur in both regions. The first subtype is referred to as the brain-first subtype, while the second is the body-first subtype. In the brain-first subtype, α-synuclein pathology mainly spreads ipsilaterally, since most brain connections are of this type. Conversely, in the body-first subtype, dissemination occurs bilaterally, since the innervation of the enteric system by the vagus and parasympathetic nerves overlaps laterally. Therefore, the DMV would be bilaterally affected, leading to subsequent dissemination, and also in a bilateral way, to other brain regions, such as the basal ganglia [26].

According to the SOC model, OD is more related to the body-first subtype because in the brain-first subtype, with the damage of only one side of the olfactory-related structures, the contralateral structures could perform the olfactory function. On the other hand, since both sides are affected in the body-first subtype, there is no possible compensation [26].

Figure 1 illustrates some of the most critical studies contributing to the comprehension of OD in PD.

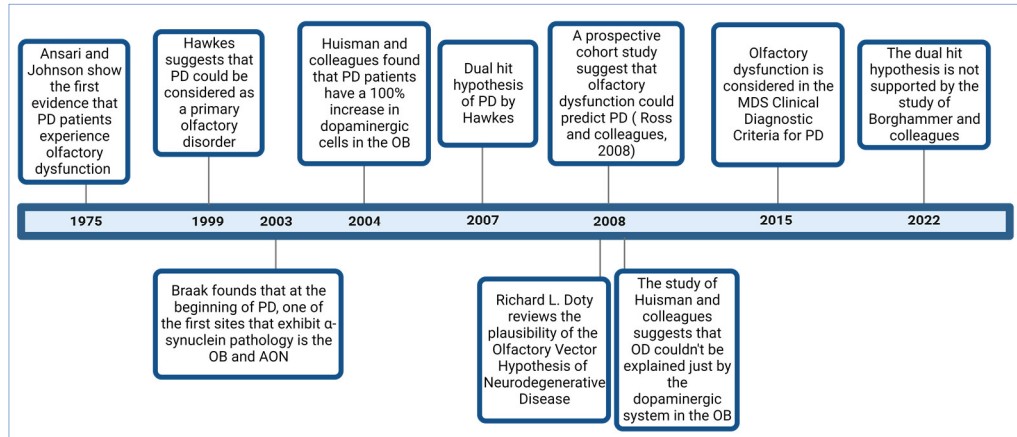

**Figure 1.** Timeline with some of the most important studies related to the comprehension of OD in PD.

## 4. Prevalence of Olfactory Dysfunction and Its Role in Diagnosis and as a Marker of PD Progression

Nowadays, it is indisputable that most patients with idiopathic and familial PD develop OD. The prevalence of this symptom was estimated in a multicenter study with patients from Germany, Australia, and the Netherlands. It was reported that, out of 400 patients with the disease, only 3.3% were normosmic. In contrast, 45.0% had anosmia, and 51.7% hyposmia. Additionally, considering the age-based norms (i.e., the olfactory capacities of people without any neuropathology according to their age group), it was found that 74.5% had some impairment of their olfactory function [15].

Additional prospective longitudinal studies have been conducted to clarify the role of OD in the prodromal period of PD. Among these studies, in 2007, a study by Haehner and colleagues followed a group of 30 patients with idiopathic hyposmia. After four years, 7% had developed IPD, so the authors considered that this sign could be the first of this disease [28]. On the other hand, in 2018, in a study by Haehner and colleagues, 474 patients with idiopathic olfactory loss were followed, and on average, 10.9 years later, 45 of them (9.8%) received a diagnosis of PD [29]. Considering the above, OD could be an early biomarker of PD and, therefore, be valuable in assisting in the early diagnosis of PD [30].

In addition, OD, in combination with other symptoms, could increase their value as early biomarkers of PD. For instance, a recent study provides evidence that individuals with isolated RBD can exhibit OD and dysprosody, a decreased pitch variation during speech (monopitch) when the nigrostriatal pathway is still intact [31].

Early diagnosis would allow for neuroprotective treatment before further cerebral involvement occurs [32], considering that by the time the motor symptoms appear, more than 60% of dopaminergic neurons in the nigrostriatal pathway have already undergone neurodegeneration [33]. Additionally, detecting OD in the early stages can be helpful in differential diagnosis, considering that OD is milder in other Parkinsonian syndromes such as drug-induced Parkinsonism, MSA, PSP, and Cortico-basal Degeneration (CBD) [30,34].

Furthermore, whether OD correlates with the progression of PD has been investigated. A review study by Fullard and colleagues suggests that OD is not a useful biomarker since no significant association has been found between these variables [30]. However, a recent systematic review that included nine longitudinal studies that followed patients for an average of 38 months found that olfactory capacity decreased over time, deteriorating more rapidly in the early stages of the disease, suggesting a possible relationship with the progression of PD [35]. On the other hand, another study showed an association between OD and a faster progression of PD [36]. Regarding gender, although a study did not find differences between men and women with PD in odor identification [37], in another study, men showed poorer olfactory performance [38].

## 5. Assessment of the Olfactory System in Patients with PD

Psychophysical tests designed to assess olfactory function allow the measurement of three main olfactory domains: odor identification, threshold, and discrimination. In identification tests, the subject must smell an odorant and then respond to what it smells like, choosing from a series of written response options. The number of correct responses is used to evaluate the subject's performance [39].

Odor identification is the olfactory domain more used for assessing OD in PD [40]. Among the most used tests for evaluating identification are The University of Pennsylvania Smell Identification Test (UPSIT), developed by Doty and colleagues, consisting of 40 items [41], and its abbreviated version of 12 items called the Brief Smell Identification Test (B-SIT). Another commonly used test is the "Sniffin' Sticks" Test, which in addition to identification, assesses threshold and discrimination. The test also yields a total score known as the TDI (Threshold, Discrimination, Identification) [42].

Threshold tests evaluate two aspects, absolute threshold and recognition threshold. The former refers to the lowest odorant concentration that a person can detect. In contrast, the recognition threshold refers to the minimum amount of the odorant that a person can identify or recognize [43]. In these tests, the participant compares the intensity of two or more stimuli, one odorant and one without odor, instead of indicating whether they perceive an odor. Recognition thresholds are measured similarly, but the subject must identify the specific smell. These forced-choice procedures are less prone to response bias [43]. For evaluating odor discrimination, it is not necessary for the participant to identify the odor, but rather to have the ability to differentiate between different odors. For example, two odoriferous stimuli are presented, and the subject has to answer as to whether they smell similar or different [39].

## 6. The Role of α-Synuclein and Its Aggregates in the Pathophysiology of PD

α-synuclein is a protein composed of 140 amino acids with a molecular weight of 14 kDa, encoded by the *SNCA* gene and located on chromosome four in humans [44,45]; it is found in axons and presynaptic terminals [46]. Evidence suggests that it is involved in functions such as transporting and filling synaptic vesicles, releasing neurotransmitters in the synapse, and plasticity mechanisms [47,48].

Although the presence of α-synuclein is not pathological, it can play a role in certain conditions known as synucleinopathies, including PD, dementia with Lewy bodies (DLB), and MSA. In these pathologies, the protein can misfold and combine with other molecules to form abnormal aggregates called Lewy bodies (LBs) that have a globular shape and are located in the cell body, or Lewy neurites (LNs), which have a thread-like shape and are found in axons or dendrites; these are both difficult to eliminate via mechanisms such as the Autophagic-Lysosomal Pathway (ALP) [20]. LBs and LNs can affect cellular functioning through processes such as the disruption of axonal transport, synaptic dysfunction, mitochondrial dysfunction, oxidative stress, endoplasmic reticulum stress, or ALP dysregulation [49,50].

Even though the loss of dopaminergic neurons in the SNc has been considered the hallmark of PD, the study by Braak and colleagues provided evidence that other brain areas are affected before this brain region. In Stage 1, α-synuclein pathology is observed in the OB and AON, as well as in the motor nuclei of the glossopharyngeal and vagus nerve. Stage 2 encompasses the medulla oblongata and pontine tegmentum. Stage 3 primarily manifests in the SNc of the midbrain. Stage 4 involves the basal forebrain and neocortex. Finally, stages 5 and 6 affect the neocortex, including sensory association cortices, the prefrontal cortex, and the premotor cortex [20]. It is estimated that the stages proposed by Braak apply to up to 80% of cases [51].

It is worth noting that Braak and colleagues indicate that the progression of α-synuclein pathology extends from two points: (1) the brainstem, with an ascending progression, and (2) from the OB and AON, to reach other structures related to the olfactory system [20].

Considering that most PD patients present with OD, it has been suggested that this symptom could be due to damage to these structures [20].

## 7. Neuroanatomical Alterations in PD Patients

To understand the mechanisms underlying OD in PD, evaluations have been carried out in vivo in patients using functional and structural neuroimaging and histological studies in postmortem patients. The following sections address some of the most relevant findings found in the structures related to the olfactory system that could contribute to olfactory loss in PD patients.

### 7.1. Olfactory Epithelium

The human olfactory system starts in the olfactory epithelium on the roof of the nasal cavity. This region contains the endings of olfactory sensory neurons (OSNs), which are bipolar neurons with cilia on the apical dendrite, where olfactory receptors are located. These receptors are G protein-coupled receptors (GPCRs) to which odorant molecules bind. The axons of the OSNs form the olfactory nerve, which crosses the cribriform plate of the ethmoid bone and synapses with second-order neurons, mitral, and tufted cells. The olfactory epithelium is pseudostratified columnar and, in addition to OSNs, contains basal cells, which are stem cells, and sustentacular or supporting cells [52].

In a study, the olfactory epithelium was evaluated to determine the presence of $\alpha$-, $\beta$- and $\gamma$-synucleins in patients with Alzheimer's disease, DLB, MSA, PD, and patients without these diseases. Among the different synucleins, $\alpha$-synuclein was the most common type in the olfactory epithelium, primarily located in OSNs and basal cells. However, patients with the diseases did not have higher levels of this protein than controls. The authors suggest that this protein could be involved in the neuroplasticity of OSNs [53].

Another study found that six out of eight patients with PD had LBs in OSNs [54]. In contrast, in the study by Witt and colleagues, no differences were found in the distribution or expression of $\alpha$-synuclein in the olfactory epithelium of PD patients compared to controls [55]. Although biopsies of the olfactory epithelium could be easily obtained and used to search for $\alpha$-synuclein pathology as a possible support for the diagnosis of PD, previous studies have suggested that due to the variability in the presence of this protein, the olfactory epithelium does not seem to be a viable option for this purpose [56].

### 7.2. Olfactory Bulb

The OB is the first station of olfactory processing. It is an extension of the telencephalon composed of six distinct layers. The layers, from outer to inner, are called the nerve layer, glomerular layer (where glomeruli localize, comprising the endings of OSNs that make synapses with the apical dendrites of mitral and tufted cells, and dendrites of interneurons called periglomerular cells), the external plexiform layer (where tufted cell bodies are found), the mitral cell layer (which consists of bodies of mitral cells), the internal plexiform layer, and the granule cell layer, consisting of granule cells [52].

One of the early findings in the research of the OB in nine PD patients was the presence of LBs in mitral cells [57]. In one study, the presence of LBs and LNs was reported in structures such as the OB, AON, and the olfactory tract before being observed in the SNc [58]. The observations of Braak and colleagues provided further support for these findings, as they documented the presence of these aggregates in the OB after stage I of their staging of PD pathology [20]. Additional evidence has confirmed these aggregates' presence in the OB [59–61]. According to Borghammer and colleagues, as reported by Braak and colleagues, Lewy pathology could start in the OB and may also reach the OB via its spread from the locus coeruleus [26].

Regarding the number of neurons in the OB, initially, Huisman and colleagues reported a 100% increase in dopaminergic neurons [21]. However, upon doubling the sample size, this difference was only observed in females rather than men [22]. In another study of six subjects with PD, more dopaminergic periglomerular neurons were observed than in the

controls [60]. On the other hand, Cave and colleagues did not find changes in the number of tyrosine hydroxylase-expressing neurons in the OB of male PD patients [62].

At a general level, without distinguishing neuronal types, a decrease in neuronal density in the OB was found in a sample of seven PD patients [63]. In another study, a significant reduction in mitral and tufted cells and Calretinin-positive interneurons was observed [62]. Additionally, in a study conducted by Zapiec and colleagues in five patients with PD and six controls, it was found that PD patients may have fewer glomeruli, which may also be smaller than controls [64].

Several studies using MRI have been carried out to determine whether there are changes in the volume of the OB in PD patients. A meta-analysis of six case–control studies suggested that the volume of both OBs in PD patients is smaller than in healthy controls [65]. Furthermore, it was found that the volume of the right OB was larger in these patients. A smaller volume was also observed in a subsequent study not included in this analysis [66]. However, in another later study using stereological methods, no significant differences were found [61].

Based on the study by Pearce and colleagues [63], Brodoehl and colleagues suggested that PD patients may have a smaller OB size due to a significant decrease in the number of neurons [67]. A recent study analyzed the possible association between α-synuclein pathology and neuronal loss in the OB. According to the results, no significant reduction in Neu-N-positive neurons was found as the α-synuclein density increased [61]. Moreover, it is worth noting that the same study showed an increase in astrogliosis and microgliosis in PD patients compared to controls, mainly in men [61].

Considering that the density of α-synuclein pathology in the OB correlates with the severity of motor symptoms and cognition, and with α-synuclein pathology in other brainstem structures, including the parietal, temporal, and frontal lobes, it has been proposed that a biopsy from the OB tissue could be helpful as a diagnostic confirmation for PD [68].

### 7.3. Anterior Olfactory Nucleus

The AON is a cortex-like structure comprising two layers. Some authors divide the AON into bulbar, retrobulbar, interpeduncular, and cortical regions [69]. Its function facilitates reciprocal information exchange from the OB to the piriform cortex, between the OBs of both hemispheres and between the respective piriform cortices through the anterior commissure [70].

In Braak's staging, at stage one, α-synuclein inclusions are found in the AON and show severe damage at stage four [20]. Additional studies have confirmed α-synuclein pathology in the AON of OD patients [61,71]. Ubeda-Bañon and colleagues found LBs and LNs in the AON's bulbar, retrobulbar, interpeduncular, and cortical regions. Specifically, these aggregates were found in cells expressing the substance P, parvalbumin, calbindin, and calretinin [69]. In addition to neurons, the role of glial cells has also been investigated in this structure. α-synuclein inclusions were found in microglia, pericytes, and astrocytes, but not in oligodendrocytes. This may be due to glial cells participating in the uptake and degradation of α-synuclein [3].

### 7.4. Olfactory Tract

The olfactory tract (OT) runs from the OB and extends posteriorly along the ventral part of the frontal lobe. It is composed of myelinated axons from mitral and tufted cells. The OT transmits information to the primary olfactory cortex, composed of the AON, olfactory tubercle, piriform cortex, anterior entorhinal cortex, peri amygdaloid cortex, and the anterior cortical nucleus of the amygdala [70].

An imaging technique used to evaluate the OT is Diffusion Tensor Imaging (DTI), which uses fractional anisotropy (FA) to measure the structural integrity of axons through fiber density, myelin structure, and axonal diameter. FA is calculated based on the movement of water molecules within the cylindrical-shaped axons, which generates an anisotropic diffusion of water molecules within the axon [72]. The FA value ranges from

0 to 1. A value of zero indicates the isotropic movement of water molecules, i.e., the cerebrospinal fluid, while a value of one represents anisotropic movement, i.e., the nerve fibers [73]. In a study involving 23 patients with PD, a lower FA and a smaller OT volume were found compared to the controls [74]. Another study utilized a high-resolution MRI sequence in combination with voxel-based statistical analysis and found a reduction in the volume of the OT in patients with PD [75].

### 7.5. Piriform Cortex

The piriform cortex is part of the primary olfactory cortex and is crucial for perceiving and discriminating odors and olfactory memory [76]. It is also responsible for processing complex mixtures of synthetic odorants and short-term olfactory habituation [76]. As part of the paleocortex, it comprises three layers. Layer I, the molecular and outermost layer, is where axon terminals from tufted and mitral cells synapse; layer II is densely populated by pyramidal neurons and semilunar cells; and layer III is composed of polymorphic cells [77,78]. An fMRI study showed that the right piriform cortex was more activated than the left while judging the familiarity of an odor [79]. Histological analysis has demonstrated the presence of LBs in the Piriform cortex [80]. Using Voxel-Based Morphometry (VBM), a smaller Gray Matter Volume (GMV) was found in the right piriform cortex of patients with PD [75]. Additional studies have found that the smaller the volume of this area (atrophy), the worse the olfactory performance [81–83].

### 7.6. Hippocampus

The hippocampus is an elongated structure that is wider in the anterior portion and becomes narrower in the posterior. It is in the medial part of the temporal lobe, adjacent to the lateral ventricles, and is approximately 4 to 4.5 cm long. It can be divided into the head, body, and tail [84]. The hippocampus receives input from the piriform cortex [85], but the primary source of information comes from the entorhinal cortex via the perforant pathway, which reaches the dentate gyrus. Then, this structure forms a synapse with the CA3 region through the mossy fibers, and from there, forms a synapse with the CA1 region through the Schaffer collateral pathway; it finally sends efferent projections to the entorhinal cortex, completing the classical trisynaptic circuit of the hippocampus [86].

Although the hippocampus is primarily known for its role in learning and memory, it also participates in the central processing of odors [87]. Together with the amygdala, it is considered part of the limbic olfactory pathway [71]. According to the Braak stages, this structure is affected by $\alpha$-synuclein pathology in stages three or four [88]. Using fMRI, it was observed that hyposmic patients with PD showed a decrease in its activation [89].

The volume of this structure was evaluated in 18 patients with PD and compared to 18 normosmic controls. A lower volume was found in both hemispheres in patients with PD and hyposmia, being more pronounced in the body subfields. Furthermore, this was correlated positively with scores obtained in an odor identification test [87]. In a study by Bohnen and colleagues, selective hyposmia was investigated, in which individuals had difficulty identifying some specific odors but not others. This study revealed selective hyposmia for dill pickle, banana, and licorice. Additionally, using Positron Emission Tomography (PET) within the same study, dopamine transporter binding was measured in the ventral and dorsal striatum, amygdala, and hippocampus, finding a correlation between the dopamine binding mainly in the hippocampus and selective hyposmia. The authors interpreted these findings as suggesting that the dopaminergic innervation of the hippocampus is implicated in the higher-level cognitive processes required for the odor identification task [88].

### 7.7. Orbitofrontal Cortex

The orbitofrontal cortex (OFC) is part of the neocortex and gets its name from its location above the eye sockets. It has been considered a secondary olfactory cortex since it participates in olfactory information processing [70] and has functions in olfactory

recognition memory [90]. In healthy subjects, it has been shown that the medial OFC cortical thickness is positively related to olfactory performance [91,92]. Specifically, the right OFC has been associated with the conscious perceptual experience of odors [93]. At the same time, the left has been implicated in evaluating odors in terms of their pleasantness or unpleasantness, also known as hedonic odor judgment [94]. Using VBM, a decrease in volume has been observed in subjects with anosmia without neurodegenerative diseases [95].

Employing electroencephalography, it was discovered that odor recognition deficits in PD are related to reduced activation in the OFC [96]. It has also been reported that a more significant loss of gray matter in this area is associated with worse olfactory function in patients with PD [81]. However, in an additional study conducted on 20 patients with PD, no association was found between these variables [97]. In a study of 24 patients with PD, DTI was used to evaluate the relationship between OD and white matter integrity through fractional anisotropy (FA) in central areas of the olfactory system. The results indicated that patients with OD had lower FA values in the OFC, particularly in the areas adjacent to the straight gyrus [98].

The study by Silveira-Moriyama and colleagues found α-synuclein pathology in the OFC in patients with PD [80]. One study shows that if α-synuclein pathology affects the OFC, it is more likely to be diagnosed as clinical PD. However, if the pathology is limited to the OB or OT, the diagnosis of this disease is less likely [71].

### 7.8. Amygdala

The amygdala, also known as the amygdaloid complex, is a brain structure located in the ventral part of the brain, specifically in the anteromedial region of the temporal lobes. It consists of different nuclei divided into three groups: basolateral, central, and corticomedial. The latter connects with the olfactory system and is considered the primary olfactory cortex because it receives monosynaptic afferents from the OB. Specifically, it establishes connections with the cortical, medial, and periamygdaloid complex nuclei [99].

In an analysis of 18 patients with PD, LBs were demonstrated in approximately 4% of neurons in the amygdala, mainly in the basolateral and cortical nuclei. In addition, a stereological analysis estimated a 20% reduction in volume. This reduction was primarily due to a 30% reduction in the volume of the corticomedial complex (central, medial, and cortical nuclei). The study suggests that the decline in the amygdala volume may be related to OD in patients with PD, as the cortical nucleus has essential connections with the OB [100].

An additional MRI study with 115 PD patients with PD and 78 healthy controls found reduced GMV. However, there were no significant differences in the volume of the amygdala nuclei [101]. By using the resting-state functional Magnetic Resonance Imaging (rsfMRI) technique, an imaging technique performed while the subject is not engaged in an explicit task, it was determined that, in patients with PD, certain regions of this structure, such as the left centromedial cortex, left and right basolateral, and left superficial, had reduced connectivity with several areas of the brain, including the olfactory cortex. The same study found a negative correlation between the severity of anosmia and the functional connectivity of the different subregions of the amygdala [101].

An additional report using rsfMRI found that patients with severe hyposmia exhibit altered functional connectivity between the amygdala and other brain regions, such as the inferior parietal lobe and the fusiform and lingual gyrus [102]. Morphometric analysis using MRI found positive correlations between olfactory performance and GMV in the right amygdala in moderately advanced PD patients [82]. Chen and colleagues found a smaller volume of gray matter in the amygdala in these patients [75]. An additional study also found a reduction in gray matter in the right amygdala [103].

*7.9. Cerebellum*

The cerebellum is in the posterior cranial fossa and comprises a medial structure called the vermis and the lateral hemispheres. Although it has traditionally been attributed to motor functions, it is also known to play cognitive and sensory roles [104]. In 1998, a study by Sobel and colleagues was the first to demonstrate the involvement of this structure in the olfactory system [105]. Using fMRI and an olfactory task, the researchers found a negative relationship between odorant concentration and airflow volume, with activation in the posterior part of the cerebellar hemispheres, specifically in the inferior semilunar lobule, superior semilunar lobule and the posterior part of the quadrangular lobule. This activation was positively correlated with the concentration of the presented odor, suggesting that the cerebellum may participate in a feedback mechanism to regulate airflow volume based on odor concentration. Additional fMRI studies found cerebellar activation in response to olfactory stimulation [106–109]. However, it should be noted that the pathway by which olfactory information reaches the cerebellum and its specific function in this sensory system has not yet been determined [109,110].

Considering that a decrease in the cortical gray matter was observed in subjects with anosmia [95,111], the cerebellum may be involved in OD in patients with pathologies primarily affecting the cerebellum. For example, it has been observed that patients with spinocerebellar ataxias exhibit OD [112,113]. In the case of PD, Sobel and colleagues showed that besides the odor identification impairment, these patients also had altered olfactomotor functions, as a significant decrease in airflow rate during sniffing was detected. The authors suggested that this alteration could be one of the reasons for OD in PD, given that sniffing, as a fine motor process, could be controlled by the cerebellum [114].

Moreover, another study showed that patients with unilateral cerebellar lesions had an impairment regarding the identification of odors using the nostril contralateral to the lesion, suggesting a contralateral connection between each nasal cavity and a cerebellar hemisphere. Additionally, they displayed deficiencies in olfactomotor abilities, as a lower volume and sniffing speed were detected compared to the controls [115].

## 8. Discussion

Prospective longitudinal studies strongly supported that OD precedes motor symptoms in PD by at least 4 years [4,29], deteriorating more rapidly in the early stages of the disease [35]. This symptom has a prevalence of up to 97% in patients with PD [15]. Therefore, it has been accepted that OD is a cardinal NMS of this neurodegenerative disorder [116].

Although OD could be a very non-specific biomarker of PD, given the numerous conditions in which there is also smell loss, such as viral infections of the nasal cavity, traumatic brain injuries, sinonasal diseases, and many neurodegenerative disorders [117,118], due to the vast accumulated scientific evidence, OD is currently a supportive criterion in the diagnosis of PD [25]. Whether this NMS is related to the disease progression is still inconclusive, but there is evidence of this relationship [30,35].

The pathophysiological mechanisms underlying olfactory loss have yet to be fully elucidated [40,102]. Nevertheless, multiple studies have been conducted to understand this condition. The evaluation of olfactory function has been studied using psychophysical tests to measure odor identification, discrimination, and threshold [40]. In order to investigate the neuroanatomical correlates that underlie OD, the brains of post-mortem patients have been examined using histological stains such as Hematoxylin and Eosin and immunostaining against α-synuclein to identify the presence of LBs and LNs in neurons and glial cells in areas linked to olfaction. Immunohistochemistry is also utilized to identify different types of cells. Using these techniques, the number of neurons has been counted, and in some cases, astrogliosis and microgliosis have been evaluated. Stereological analyses have also been performed to measure the volume of some structures related to the olfactory system.

Neuroimaging techniques such as MRI have been used in living patients to perform morphometric analyses using VBM. Additionally, DTI has been used for white matter

analysis, evaluating parameters such as fractional anisotropy and radial, medial, and axial diffusivity [74]. Similarly, fMRI has been used to study the functionality of some brain areas, as well as rsfMRI. PET is another neuroimaging technique employed to evaluate dopaminergic innervation. Furthermore, EEG has been used to assess brain electrical activity [96]. Through all these techniques, the findings described below have been found.

Regarding the olfactory epithelium, two studies did not find differences in the presence of α-synuclein pathology between PD patients and controls [53,55]. However, a later study showed its presence [54]. It should be noted that one of the limitations of these studies is their small sample size. Furthermore, no specific characteristics have been identified in the olfactory epithelium of PD patients that differ from those with hyposmia due to other causes, such as post-infectious olfactory loss or congenital anosmia [119]. Therefore, alterations in olfactory function in PD patients could be more related to abnormalities in the central olfactory structures of the brain [119].

An interesting finding is that some olfactory receptors are expressed in various parts of the nervous system beyond the OSNs of the olfactory epithelium, including the frontal cortex. Although a decrease in the gene expression of these receptors has been observed in the frontal cortex of PD patients, there is no evidence establishing a direct relationship between these changes and OD in PD [120].

Among all the structures involved in olfaction, the OB is one of the most studied in PD patients. Braak and colleagues reported that, along with the AON, these structures were the first sites affected by α-synuclein pathology [20]. Several studies have also identified the presence of LBs and LNs in the OB [22,58–61], which precedes the clinical symptoms of PD [71]. Additionally, the OB is a relevant structure in the olfactory vector hypothesis [9] and the dual-hit hypothesis of PD [23]. In the OB, a decrease in mitral and tufted neurons and Calretinin-positive interneurons has been found [62], while the number of dopaminergic neurons is increased significantly in women [22,62]. Additionally, reductions in the glomerular number and size have been found [64]. Furthermore, astrogliosis and microgliosis have been reported [61,121].

On the other hand, MRI studies have not been entirely consistent, but a systematic review and meta-analysis have suggested a decrease in the OB volume in these patients [65]. The vector hypothesis for PD indicates that the OB may be affected by an agent, such as a toxin or virus, that enters the brain through the nasal cavities and causes LBs and LNs. Another possibility is that α-synuclein pathology can occur due to the dissemination from the locus coeruleus [122].

The AON presents α-synuclein pathology in parallel with the OB [20,61]. In the AON, additional studies have found that LBs and LNs are present in neurons expressing substance P, parvalbumin, calbindin, and calretinin [69]. Stevenson and colleagues also reported α-synuclein pathology in non-neuronal cells such as microglia, pericytes, and astrocytes in this structure [3]. On the other hand, in the OT, using DTI, a decrease in its volume and a significant fractional anisotropy indicates changes in its integrity [74]. Considering the reduction in the number of mitral/tufted cells [62], this could contribute to the changes observed in this structure.

Despite being part of the primary olfactory cortex, the piriform cortex has been little studied. Using VBM, a change in the volume of patients with PD has been reported, which correlates to OD [75,81–83]. Regarding the hippocampus, α-synuclein pathology has been found in this region in PD patients [71]. In addition, it was observed that these patients with hyposmia have reduced activation [89]. Likewise, using VBM, it was found that PD patients with hyposmia had smaller volumes, specifically in the portion called the body [87]. Additionally, a correlation was found between the dopaminergic innervation of this brain structure and selective hyposmia to certain odors [88].

The OFC has been implicated in odor identification [97,123]. In PD patients, an association was identified between the deficit in odor identification and the decrease in its activation [96]. The loss of gray matter in this area is positively correlated with olfactory function in PD patients [81]. It is important to note that besides its olfactory functions, the

OFC is also involved in cognitive processes [124]. On the other hand, OD correlates with cognition and may predict the progression to dementia in PD patients [125]. Therefore, the impairment in the OFC may interfere with odor identification.

It is important to note that OD in PD relates to $\alpha$-synuclein pathology in neocortical areas more than in the OB [26,71,126]. Additionally, most tests used in olfactory evaluation measure identification, which could involve cognitive processes such as episodic and semantic memory [123,127]. This could explain why the body-first subtype cases are more prone to developing OD as, in this subtype, some cortical structures that perform high-order olfactory functions, including the OFC, are affected [26].

Regarding the amygdala, it has been found by rsfMRI that its subregions have altered functional connectivity with the olfactory cortex and other brain areas [101,102]. In this region, $\alpha$-synuclein pathology has also been found mainly in the basolateral and cortical nuclei, which have essential connections with areas involved in the olfactory system [100]. Regarding its volume, there is no consensus as to whether changes occur, as in some cases, it has been seen to decrease, while in others, not [100,101]. In addition, gray matter atrophy in the right amygdala positively correlates with olfactory functions in moderately advanced PD patients [82].

As for the cerebellum, several studies have suggested that it could also be involved in the olfactory system [105–109]. However, the pathway by which olfactory information reaches this structure and its specific function are still unknown [109,110]. Nonetheless, authors such as Sobel and colleagues found that patients with PD present a dysfunction in sniffing, a fine motor activity in which the cerebellum could be involved [114]. Therefore, this condition could contribute to the mechanisms of OD.

Given the potential role of $\alpha$-synuclein pathology in the atrophy of olfactory-related structures and considering all the effects the LBs and LNs could have on the cells [49,50], a study found that increasing the $\alpha$-synuclein density did not result in a significant decrease in Neu-N-positive neurons [61]. In addition, in patients with DLB who underwent MRI, a negative correlation was observed between the amygdala volume and the presence of LBs. Still, no such correlation was found for the hippocampus and entorhinal cortex [128].

Considering that $\alpha$-synuclein pathology has been observed in most structures associated with the olfactory system, it has been suggested that this could account for the OD in patients with PD [20,129]. Therefore, a relationship between the presence of these aggregates and OD could be expected. Ross and colleagues demonstrated an association between the presence of these aggregates in the SNc and locus coeruleus and OD [130]. In another study, individuals with LB in the brain's neocortical and limbic regions demonstrated a poorer performance in the odor identification test, even if they did not present symptoms of dementia or PD [131].

Although Braak and colleagues demonstrated the progression of the $\alpha$-synuclein pathology in the brain, it remains unclear how this pathology spreads from the OB and AON to other brain regions associated with the olfactory system [20]. In one study, $\alpha$-synuclein fibrils were infused in the OB and AON in mice. After three months, inclusions with phosphorylated $\alpha$-synuclein were observed in the hippocampal formation, amygdala, and piriform and entorhinal cortices, but not in areas not directly connected to the OB and AON. The study suggests that the $\alpha$-synuclein pathology could have spread retrogradely through neuroanatomical connections [129]. Despite these findings not being directly applicable to humans, considering the denser connectivity between these brain structures in rodents, it is a good advancement in our understanding of the spreading of the $\alpha$-synuclein pathology to the olfactory structures.

It is important to mention that many other structures not reviewed in this article could be involved in the pathophysiology of OD in PD. For instance, in PD patients with severe hyposmia, a reduction in GMV in the right associative visual area and inferior, middle, and superior frontal gyri has been reported [102]. The neurotransmitter systems, mainly dopamine, serotonin, acetylcholine, and norepinephrine, also could be implicated.

In summary, OD is one of the most common NMSs of PD. Since it precedes a diagnosis by at least four years, it could be considered a potential biomarker for this neurodegenerative disease. Although the mechanisms underlying OD in PD are not fully understood, certain functional and structural changes have been identified both in the peripheric structures and the primary and secondary olfactory cortices. The olfactory epithelium does not show specific differences between PD patients and controls or those with OD from other causes. This suggests the involvement of central olfactory structures in the development of OD. The OB and AON are some of the first affected brain structures in PD, showing the presence of $\alpha$-synuclein pathology. Furthermore, the OT exhibits a reduction in its volume and a change in its integrity, as determined by fractional anisotropy.

Other brain regions such as the piriform cortex, hippocampus, OFC, and amygdala also exhibit $\alpha$-synuclein pathology and alterations in volume, activation, and connectivity. Additionally, growing evidence suggests that the cerebellum participates in the olfactory system and since PD patients present alterations in sniffing, this brain structure could be implicated in this alteration and thus in the pathophysiology of OD in PD. Considering the multitude of structures associated with the olfactory system affected in PD, it is likely that a complex network of interactions underlies the mechanisms by which PD patients experience OD.

## 9. Future Directions

Some brain structures involved in the olfactory system, such as the OB, amygdala, and hippocampus, have been more widely studied in PD patients with OD. However, other regions have received less attention, including the piriform cortex, so there is a need for further investigation into these brain structures. Furthermore, since certain types of neurons are more susceptible to developing $\alpha$-synuclein pathology, it is necessary to identify which types of cells are more affected in these structures. It is worth noting that glial cells have not been extensively studied except for a few cases.

In addition to the cardinal motor symptoms of PD, patients also exhibit an impairment in sniffing, which could contribute to OD. This fine motor impairment may implicate the cerebellum. Therefore, further studies on sniffing in PD patients and the role of the cerebellum in OD are necessary.

Considering that the assessment of the olfactory function is simple, non-invasive, and cost-effective, it is ideal for being performed on those individuals already at risk of PD, such as those patients with family members with clinical PD or other non-motor symptoms characteristic of this neurodegenerative disease. Additionally, it is crucial to perform this type of evaluation in patients already diagnosed with PD in order to understand the pathophysiology of this symptom.

## 10. Conclusions

Scientific evidence shows functional, microstructural, tissue, and morphological alterations in the structures related to the olfactory system, both peripheral and in the primary and secondary olfactory cortex. Furthermore, considering the multiple studies that show alterations in white matter and functional connectivity, and based on the recently proposed SOC model, it appears that a complex network of structures related to the olfactory system may be involved in the pathophysiology of OD in PD.

**Author Contributions:** Research and literature compilation: G.T.-P.; Critical analysis of the studies included: D.C.-C. and G.T.-P.; Information synthesis: L.I.G. and G.T.-P.; Writing and editing: L.I.G., D.C.-C., P.C.-C., G.M., M.E.H.-A., G.E.A.-A., J.M. and G.T.-P. All authors have read and agreed to the published version of the manuscript.

**Funding:** This research received no external funding.

**Institutional Review Board Statement:** Not applicable.

**Informed Consent Statement:** Not applicable.

**Data Availability Statement:** Not applicable.

**Acknowledgments:** This work was made possible by the support of CONACYT, who granted a scholarship to Torres-Pasillas, G. (scholarship number 926589), allowing for the successful completion of this research paper.

**Conflicts of Interest:** All authors declare that they have no conflict of interest in its publication.

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
