# Peer review of "Olfactory Dysfunction in Parkinson’s Disease, Its Functional and Neuroanatomical Correlates"

_neurosci, doi:10.3390/neurosci4020013_

Round 1
Reviewer 1 Report
thank you for asking me to review this very comprehensive and clearly written review on Olfactory dysfunction in PD.
The authors have made an effort to write exhaustively about the subject. It does read more like a Book chapter than an article.
Could the authors summaries the main points in the discussion?
Could they also refer to the study by Rusz Jan et al (2022) that links olfactory dysfunction and dysprosody (in Movement Disorders) and then move to a more recent bibliography, in the area of biomarkers.
Author Response
Thank you for recognizing the quality of our review on olfactory dysfunction in Parkinson's disease.
The main points have been summarized in the final part of the discussion section.
The article titled “Dysprosody in Isolated REM Sleep Behavior Disorder with Impaired Olfaction but Intact Nigrostriatal Pathway,” by published in the Movement Disorders Journal, was added in the section of the manuscript called “4. Prevalence of olfactory dysfunction and its role in diagnosis and as a marker of PD progression”. This scientific article further supports the potential of combining olfactory dysfunction with other symptoms as biomarkers for Parkinson's disease.Reviewer 2 Report
Dear Authors,
the Manuscript entitled “Olfactory dysfunction in Parkinson's disease, its functional and neuroanatomical correlates” is a Review that aim to evaluate the olfactory dysfunction in Parkinson's disease, focusing on several portion of the central nervous system.
Abstract: the abstract need to be changed in order to unify it to the development of the manuscript.
Discussion: Authors cited numerous data from literature, some of them were cited also in the main text, but Authors have not developed a real discussion on them.
Specific Comments
Line 22: Change “Alpha-synuclein” with “α-synuclein”.
Line 60: Please don’t write “and” in italic.
Line 132: Change “α-syn” with “α-synuclein”
Line 180: Insert a space between variables and [30]
Line 214: Change “Alpha-synuclein” with “α-synuclein”.
Line 215: please format “[43]; it”
Line 232: Please change the sentence to avoid numerous “:”
Line 242: insert a line before point 7
All the manuscript: change citation references as follow: “Witt et al.” in “Witt and Colleagues”
Line 386: change “correlated” with “was correlated”
Lines 415, 416, 540, 570: Change “α-syn” with “α-synuclein”
Line 436: Change “there was no significant difference” with “there were no significant differences”
Line 497: please avoid “(H&E)”, it is not necessary
Line 636: change “α-Synuclein Origin Site and Connectome (SOC) model” with “SOC model”.
Minor editing of English language required
Author Response
The abstract was modified to mention the sections reviewed in the article and ensure coherence with the development of the manuscript.
In our discussion, the findings of the reviewed studies are summarized. Additionally, a contrast is made between the results obtained, for example, mentioning studies where a decrease in the volume of the structures has been found. In contrast, in others, no relevant variations are observed. Another example is the variability in the findings related to the increase or decrease in the number of neurons.
Furthermore, certain gaps in knowledge on the subject are identified. For instance, the complete understanding of how the propagation of α-synuclein pathology occurs in structures related to the olfactory system is still not fully comprehended.
However, it is important to note that the current scientific literature does not yet provide a precise explanation for olfactory dysfunction in patients with Parkinson's disease. In order to establish a theoretical framework for understanding this non-motor symptom of Parkinson's disease, proposals such as the classification of sporadic Parkinson's disease-related brain pathology, the olfactory vector hypothesis of neurodegenerative disease, the dual-hit hypothesis in Parkinson's disease, and the α-synuclein Origin site and Connectome (SOC) model have been considered. These theoretical approaches provide a valuable reference for addressing olfactory dysfunction in Parkinson's disease.
Additionally, specific points have been corrected.Reviewer 3 Report
The review “Olfactory dysfunction in Parkinson's disease, its functional and 2 neuroanatomical correlates” by Torres-Pasillas et al. is well written and the review will be of interest to NeuroSci readers. The authors conducted a comprehensive review of the functional and neuroanatomical alterations in Parkinson´s disease providing an extended view of the influence of odor dysfunction in this neurodegenerative disease.
Author Response
Thank you for recognizing the quality of our review on olfactory dysfunction in Parkinson's disease. We aimed to provide an in-depth understanding of this condition's functional and neuroanatomical aspects. Your positive feedback is greatly appreciated and encourages us to continue contributing to the field of Neuroscience.